# Grid Cell-Inspired Fragmentation and Recall for Efficient Map Building

**Jaedong Hwang**                                                    *jdhwang@mit.edu*
*Massachusetts Institute of Technology*

**Zhang-Wei Hong**                                                    *zwhong@mit.edu*
*Massachusetts Institute of Technology*

**Eric Chen**                                                    *ericrc@mit.edu*
*Massachusetts Institute of Technology*

**Akhilan Boopathy**                                                    *akhilan@mit.edu*
*Massachusetts Institute of Technology*

**Pulkit Agrawal**                                                    *pulkitag@mit.edu*
*Massachusetts Institute of Technology*

**Ila Fiete**                                                    *fiete@mit.edu*
*Massachusetts Institute of Technology*

**Reviewed on OpenReview:** *https://openreview.net/forum?id=cT8oOJ6Q6F*

## Abstract

Animals and robots navigate through environments by building and refining maps of space. These maps enable functions including navigation back to home, planning, search and foraging. Here, we use observations from neuroscience, specifically the observed fragmentation of grid cell map in compartmentalized spaces, to propose and apply the concept of *Fragmentation-and-Recall* (FARMap) in the mapping of large spaces. Agents solve the mapping problem by building local maps via a surprisal-based clustering of space, which they use to set subgoals for spatial exploration. Agents build and use a local map to predict their observations; high surprisal leads to a "fragmentation event" that truncates the local map. At these events, the recent local map is placed into long-term memory (LTM) and a different local map is initialized. If observations at a fracture point match observations in one of the stored local maps, that map is recalled (and thus reused) from LTM. The fragmentation points induce a natural online clustering of the larger space, forming a set of intrinsic potential subgoals that are stored in LTM as a topological graph. Agents choose their next subgoal from the set of near and far potential subgoals from within the current local map or LTM, respectively. Thus, local maps guide exploration locally, while LTM promotes global exploration. We demonstrate that FARMap replicates the fragmentation points observed in animal studies. We evaluate FARMap on complex procedurally-generated spatial environments and realistic simulations to demonstrate that this mapping strategy much more rapidly covers the environment (number of agent steps and wall clock time) and is more efficient in active memory usage, without loss of performance.[1]

---

[1] https://jd730.github.io/projects/FARMap

## 1 Introduction

Human episodic memory breaks our continuous experience of the world into episodes or fragments that are divided by event boundaries corresponding to large changes of place, context, affordances, and perceptual inputs (Baldassano et al., 2017; Ezzyat & Davachi, 2011; Newtson & Engquist, 1976; Richmond & Zacks, 2017; Swallow et al., 2009; Zacks & Swallow, 2007). The episodic nature of memory is a core component of how we construct models of the world. It has been conjectured that episodic memory makes it easier to perform memory retrieval, and to use the retrieved memories in chunks that are relevant to the current context. These observations suggest a certain locality or fragmented nature in how we model the world.

Chunking of experience has been shown to play a key role in perception, planning, learning and cognition in humans and animals (De Groot, 1946; Egan & Schwartz, 1979; Gobet & Simon, 1998; Gobet et al., 2001; Simon, 1974). In the hippocampus, place cells appear to chunk spatial information by defining separate maps when there has been a sufficiently large change in context or in other non-spatial or spatial variables, through a process called *remapping*; see Colgin et al. (2008); Fyhn et al. (2007). Grid and place cells in the hippocampal formation have also been shown to *fragment* their representations when the external world or their own behaviors have changed only gradually rather than discontinuously in the same environment (Carpenter et al., 2015; Derdikman et al., 2009; Low et al., 2021) (Figure 1a).

Inspired by the concept of online *fragmentation* and *recall (remapping to the existing fragment)* proposed for grid cells Klukas et al. (2021), we propose a new framework for map-building, FARMap, schematized in Figure 1b. This model combines three ideas: 1) when faced with a complex world, it can be more efficient to build and combine multiple (and implicitly simpler) local models than to build a single global (and implicitly complex) model, 2) boundaries between local models should occur when a local model ceases to be predictive, and 3) the local model boundaries define natural subgoals, which can guide more efficient hierarchical exploration.

As an agent explores, it predicts its next observation. Based on a measure of surprisal between its observation and prediction, there can be a *fragmentation* event, at which point the agent writes the current model into long-term memory (LTM) and initiates a new local model. While exploring the space, the agent consults its LTM, and *recalls* an existing model if it returns to the corresponding space. The agent uses its current local model to act locally, and its LTM to act more globally. We apply this concept to solve the spatial map building problem.

We first simulate animal studies (Derdikman et al., 2009; Carpenter et al., 2015) using FARMap showing its ability to fragment environments at the same locations as observed in animals. This confirms that FARMap accurately replicates the fragmentation points noted in animal research. We then evaluate the proposed framework on procedurally-generated spatial environments. Experimental results support the effectiveness of the proposed framework; FARMap explores the spatial environment with much less memory and computation time than its baseline by large margins as the agent only refers to the local model and uses both memories for setting subgoals.

The contribution of this paper is three-fold:

- We propose a new framework for mapping based on **F**ragmentation-**A**nd-**R**ecall, or **FAR**Map, that exploits grid cell-like map fragmentation via surprisal combined with a long-term memory to perform efficient online map building.

- We contribute procedurally-generated environments for spatial exploration, with parametrically controllable complex shapes that include multiple rooms and pathways.

- We demonstrate the efficacy of our framework in spatial map-building tasks. Our experiments show that FARMap reduces wall-clock time and the number of steps (actions) taken to map large spaces, and requires smaller online memory size relative to baselines.

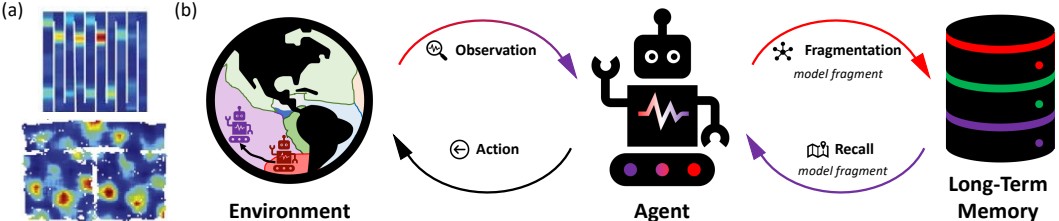

Figure 1: (a) Firing fields of grid cells in various environments from Derdikman et al. (2009) (top) and Carpenter et al. (2015) (bottom). The firing pattern changes at the boundary between two regions (fragmentation). (b) Overview of our approach. Given an observation from the environment, the FARMap agent decides whether to fragment the space based on how well it can predict the observation. If *fragmentation* occurs, the current map (or model) fragment is stored in long-term memory (LTM); the agent then initializes a new map (or model) fragment. Conversely, if the current observation closely matches the observations stored in LTM, the agent loads an existing map (or model) fragment from there (*recall*). Based on the current fragment, the agent selects an action to explore the environment.

## 2 Related Work

### 2.1 Fragmentation of Grid Cell Maps

Mammalian entorhinal grid cells generate highly regular periodic spatial representations that tile open environments (Hafting et al., 2005). This periodic response is hypothesized to be a general allothetic spatial coordinate system that represents displacements. The spatial response is independent of the speed and direction of movement and is believed to be formed through integration of self-velocity estimates. However, the regular periodic firing pattern of grid cells becomes fragmented in more complex spatial layouts, such as when an environment contains multiple subdivisions (Carpenter et al., 2015; Derdikman et al., 2009; Fyhn et al., 2007). For instance, there is a fracture in the periodic response at sharp turns of a narrow corridor and in doorways, where the grid phase appears to be remapped or jumps discretely to a distinct value. A recent manuscript (Klukas et al., 2021) builds a model to predict when such discrete remapping events might occur even though the agent explores the environment in a continuous trajectory. They formulated map fragmentation as a clustering computation, and showed how online clustering based on observational surprise results in fragmentations that match the neuroscientific observations in grid cells and also match normative clustering algorithms like DBSCAN (Ester et al., 1996). However, that work did not extensively explore the functional utility of grid cell-like map fragmentation. Here, we show that surprisal-based fragmentation, which fits the biological fragmentation data, is a biologically plausible principle that enables agents to efficiently build maps of various environments online without getting stuck in local loops.

### 2.2 Grid Cell-Inspired SLAM

Grid cells have received attention in robotics due to their potential to produce more robust spatial navigation. Milford et al. (2004) propose a model based on continuous attractor dynamics (Samsonovich & McNaughton, 1997) and more recently with grid cells (Ball et al., 2013; Milford et al., 2010), to achieve correct loop closure during noisy odometry. Similarly, Zhang et al. (2021) employ growing self-organizing maps inspired by the hippocampus for the same purpose. Yu et al. (2019) extend OpenRatSLAM (Ball et al., 2013) to 3D environments via conjunctive pose cell model that employs 3D grid cell. These methods focus on the error-correcting properties of grid cell dynamics. They do not consider fragmented grid cell maps and the possibility that these map fragments might represent the construction of subgoals which could be used for further spatial exploration.

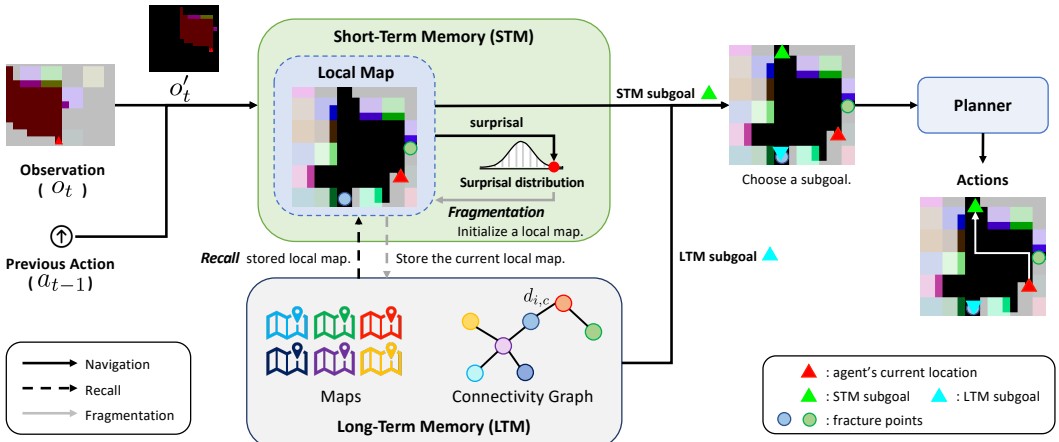

Figure 2: Illustration of the FARMap framework. Navigation (black arrow): Given the current observation which is an egocentric top-down view with a restricted field of view and previous action, the agent updates its short-term memory (STM) and selects a subgoal from the current local map in STM or the local map connectivity graph stored in LTM. The planner generates a sequence of actions for the shortest path to the subgoal. Recall (dashed arrow): If the agent arrives at a fracture point (circle in the map), a corresponding local map is recalled from LTM and the current local map stored in LTM is updated. Fragmentation (gray arrow): If the current surprisal is higher than a threshold, the current local map is stored in LTM and a new local map is initialized. $o'_t$ is a spatially transformed observation with the same size as the current local map to update the map.

## 2.3 Frontier-based SLAM

Active SLAM (simultaneous localization and mapping) agents must efficiently explore spaces to build maps. A standard approach is to define the *frontier* between observed and unobserved regions of a 2D environment, and then select exploratory goal locations from the set of frontier states (Yamauchi, 1997). Frontier-based exploration has been extended to 3D environments (Dai et al., 2020; Dornhege & Kleiner, 2011) and used as a building block of more sophisticated exploration strategies (Stachniss et al., 2004). Although conceptually simple, frontier-based exploration can be quite effective compared to more sophisticated decision-theoretic exploration (Holz et al., 2010). A cost of frontier-based exploration is the use of global maps and global frontiers, making the process memory expensive and search intensive. In contrast to frontier-based exploration, our approach *learns* the surprising parts of an environment as intrinsic subgoals, selecting among those as the exploratory goals.

## 2.4 Submap-Based SLAM

Submap-Based SLAM algorithms involve mapping a space by breaking it into local submaps that are connected to one another via a topological graph. Such Submap-Based SLAM methods are usually designed to avoid path integration errors when building maps of large spaces (*e.g.* Fairfield et al. (2010)) and to reduce the computational cost of planning paths between a start and target position (Fairfield et al., 2010; Maffei et al., 2013). Maffei et al. (2013) add DP-SLAM (Eliazar & Parr, 2003) to SegSLAM to reduce the search space, generating segments periodically at fixed time-intervals. Choset & Nagatani (2001) generate new landmarks in an environment to build a topological graph of the landmarks and navigates based on the graph. FARMap is closely related to these methods in that we build multiple submaps. However, FARMap divides space based on properties of the space (how predictable the space is based on the local map or model), and does so in an online manner using surprisal. As we show below, this fragmentation strategy can lead to improvements in performance compared to random or periodic fragmentation.

## 3 Fragmentaion and Recall based Spatial Mapping (FARMap)

### 3.1 Motivation and Overview

Animals explore spaces efficiently even in large environments by using grid cells' remapping that divides an environment into multiple subregions. This remapping can be modeled as surprisal-based online fragmentation (Klukas et al., 2021). Here, we propose a fragmentation-and-recall based spatial map-building strategy (*FARMap*) inspired by remapping of grid cells. FARMap tackles the problem of SLAM algorithms: the memory cost and search cost of finding subgoals grow rapidly with environment size; for agents exploring a large space, the computational costs could explode.

While exploring an environment, an agent builds a local model (map) and uses it in short-term memory (STM) to compute a surprisal signal that depends on the current observation and the agent's local model-based prediction. When the surprisal exceeds some threshold, this corresponds to a *fragmentation* event. At this event, the local model is written to long-term memory (LTM) which builds a connectivity graph that relates model fragments to each other so that it can share information across local models without direct access to the stored models in LTM. Then, the agent initializes an entirely new local model. Conversely, if the agent revisits the fracture point, the agent *recalls* the corresponding model fragment (local model). Hence, the agent can preserve and reuse previously acquired information. Figure 2 shows how an agent decides its next subgoal given the observation and the previous action with fragmentation and recall. LTM (except the connectivity graph portion) can be regarded as external memory, while STM is modeled as working memory. This external memory is accessed or updated only during fragmentation or recall processes. Consequently, this can be beneficial for machines with limited memory access (see Appendix E). We also discuss LTM retrieval overhead in Appendix C.

### 3.2 Overall Procedure of Spatial Navigation

Algorithm 1 presents the overall procedure of FARMap at time $t$. On top of the Frontier algorithm (Yamauchi, 1997), we have colored the FARMap algorithm blue. Given the previous action $a_{t-1}$, current observation $o_t$, a local predictive map $\mathbf{M}_{t-1}^{\text{curr}}$ at time $t-1$, we first update the map following Eq. 1 and calculate the surprisal $s_t$ following Eq. 2.

If the agent is located at the fracture point where fragmentation happened between the current local map, $\mathbf{M}_t^{\text{curr}}$ and another local map stored in LTM (Line 6), we store $\mathbf{M}_t^{\text{curr}}$ and $q_c$ in LTM, and the stored map fragment is recalled to STM. On the other hand, if the $z$-scored surprisal $z_t$ calculated using the running mean and standard deviation of surprisal within the current local map is greater than a threshold, $\rho$ (Line 9), we store $\mathbf{M}_t^{\text{curr}}$, and $q_c$ in LTM, and initialize a new map in STM. During this process, the current locations in both $\mathbf{M}_t^{\text{curr}}$ and the new map are marked as fracture points (Section 3.4). After checking recall and fragmentation, we find the desirable local map fragments that are less explored than other fragments (Section 3.6). If the current map is not the desirable map, we set the subgoal as the fracture point between the current map and the desirable map. Otherwise, we first find frontier-edges and calculate the weight of each frontier-edge $\mathcal{F}_i$ using weighted sampling with weight $w_i$ following Eq. 3 ($w_i$ is $1/d_i$ in the case of the Frontier model). The subgoal is defined as the nearest frontier from the centroid of the sampled frontier-edge. Finally, a planner generates a sequence of actions to navigate to the subgoal (Section 3.7). Note that while the agent moves based on the sequence, it keeps updating the map and checking fragmentation and recall.

### 3.3 Local Map

The STM has a local predictive spatial map, $\mathbf{M}_t^{\text{cur}} \in \mathbb{R}^{(C+1) \times H \times W}$ where height $H$ and width $W$ grow as the agent extends its observations in the local region by adding newly discovered regions. The first $C$ channels of $\mathbf{M}_t^{\text{cur}}$ denote color and the last channel denotes the agent's confidence in each spatial cell. In this paper, we will focus only on the update of the confidence channel (the $C$-th channel). The local predictive map is simply a temporally decaying trace of recent sensory observations like a natural agent (Zhang et al., 2005):

$$\mathbf{M}_{t,C}^{\text{cur}} = \gamma \cdot \mathbf{M}_{t-1,C}^{\text{cur}} + (1-\gamma) \cdot o_{t,C}', \tag{1}$$

---

**Algorithm 1** FARMap Procedure at time $t$. FARMap algorithm is colored in blue on top of Frontier algorithm (Yamauchi, 1997).

---

**Require:** a spatial map $\mathbf{M}_{t-1}^{\text{curr}}$, previous action $a_{t-1}$, current observation $o_t$, short-term memory STM, long-term memory LTM, position at time $t$, $\text{pos}_t$, decay factor $\gamma$, fragmentation threhsold $\rho$ and hyperparameter $\epsilon$.

**Ensure:** Updated map, $\mathbf{M}_t^{\text{curr}}$ and a sequence of actions $\{a\}$

  1: **procedure** STEP

Sec. 3.3 ⊰   2:      $\mathbf{M}_t^{\text{curr}} = \gamma \cdot \mathbf{M}_{t-1}^{\text{curr}} + (1-\gamma) \cdot o_t'$                 ▷ Update the current local map

  3:      Calculate $s_t = 1 - c_t$ following Eq. 2.

  4:      $z_t = (s_t - \mu_t)/\sigma_t$

  5:      $q_c = N_{\text{frontier}} \, / \, N_{\text{known}}$

  6:      **if** $\text{pos}_t =$ fracture point **then**                ▷ **Recall**

  7:        $\text{LTM} \leftarrow \text{Store}(\text{pos}_t, q_c, \mathbf{M}_t^{\text{curr}})$           ▷ Store $\mathbf{M}_t^{\text{curr}}$

Sec. 3.4   8:        $\text{STM} \leftarrow \text{Recall}(\text{pos}_t; \text{LTM})$           ▷ change $\mathbf{M}_t^{\text{curr}}$

  9:      **else if** $z_t > \rho$ **then**               ▷ **Fragmentation**

10:        $\text{LTM} \leftarrow \text{Store}(\text{pos}_t, q_c, \mathbf{M}_t^{\text{curr}})$

11:        Initialize a new map $\mathbf{M}_t^{\text{curr}}$ in STM.

12:      **end if**

Sec. 3.5 ⊰ 13:      Update running mean $\mu_{t+1}$ and standard deviation $\sigma_{t+1}$ of surprisal.

14:      $g = \arg \max_i \frac{q_i}{d_{i,c}+\epsilon}$                    ▷ Eq. 4

15:      **if** $g \neq c$ **then**         ▷ Subgoal based on connectivity between fragments.

16:        $subgoal \leftarrow$ the fracture point between the current fragment $c$ and a fragment $g$

17:      **else**

Sec. 3.6 18:        Find frontier-edges $\{\mathcal{F}_i\}$ and their centroids $\{\text{centroid}_i\}$.

19:        $d_i = ||\text{pos}_t - \text{centroid}_i||_1$.

20:        $w_i = 1/d_i \cdot |\mathcal{F}_i| \cdot \mathbb{1}(\mathcal{F}_i$ is not located spatially behind the agent$)$

21:                          ▷ $\mathbb{1}(\cdot)$ is 1 if the condition is true else 0.

22:        Select frontier-edge $\mathcal{F}_g$ based on the weighted sampling with $\{w_i\}$.

23:        $subgoal \leftarrow$ the nearest frontier $\in \mathcal{F}_g$ from its centroid.

24:      **end if**

Sec. 3.7 ⊰ 25:      A sequence of actions, $\{a\} \leftarrow \text{Planner}(subgoal; \mathbf{M}_t^{\text{curr}})$       ▷ Dijkstra's algorithm

26: **end procedure**

---

where $\gamma$ is a decay factor and $o_t \in \mathbb{R}^{(C+1) \times h \times w}$ is the egocentric view input observation in the environment at time $t$ sized as $(h, w)$. The last channel of the observation indicates visibility caused by occlusion or restricted field of view (FOV); visible (1) or invisible (0) in each cell. The red region is visible and others are invisible in Figure 2. $o_t' \in \mathbb{R}^{(C+1) \times H \times W}$ denotes a spatially transformed observation to $\mathbf{M}_{t-1}^{\text{cur}}$ to update the current observation to the local map in the correct position using rotation and zero-padding. Figure 3 shows a toy illustration of how to transform the current observation to update the local map and how the map size grows. We first rotate the observation based on the head direction of the agent in the map and then zero-pad it so that it has the same size as the local map considering the agent's current location in the map. If the observation does not fit within the map due to the agent's location, we add zero-padding (gray in the figure) to both the transformed observation and the local map. Then, we update the local map by adding the transformed observation. For example, once the exploration has started, the memory size is $h \times w$ $(H = h, W = w)$ and if the agent moves one step upward, the size changes to $(h+1) \times w$ $(H = h+1, W = w)$.

### 3.4 Fragmentation and Recall

**Fragmentation**   Fragmentation occurs if the $z$-scored current surprisal $((s_t - \mu_t)/\sigma_t)$ exceeds a threshold, $\rho$, where $s_t$ denotes surprisal at time $t$, and $\mu_t$ and $\sigma_t$ represent its running mean and standard deviation. Initially, for each new map, the agent collects surprisal statistics and is not permitted to further fragment space until the number of samples is greater than 25 (to ensure large enough sample conditions for statistics).

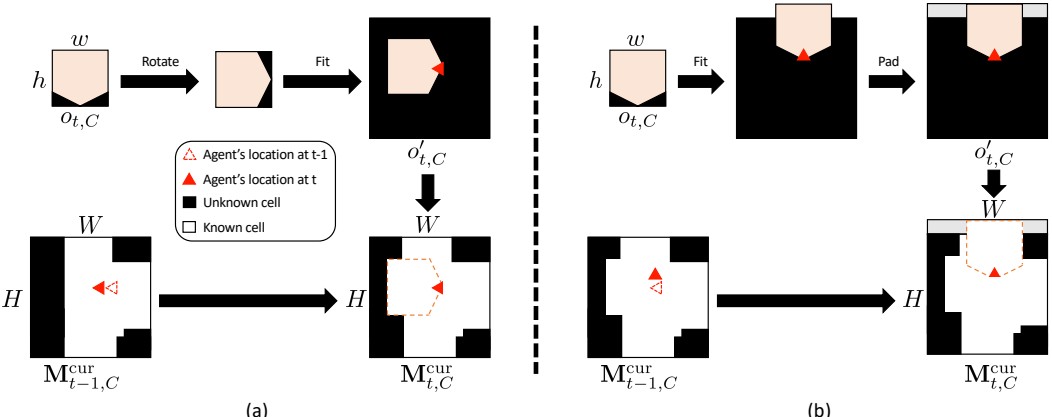

Figure 3: Schematic illustrations of how the local map is updated. In this figure, we only consider the visibility of each cell ignoring occupancy and color for simplification. (a) We first rotate the current observation $o_{t,C}$ based on the head direction of the agent in the local map. Then, the observation is zero-padded to match the same size as the local map. Finally, the local map is updated by adding the transformed observation $o'_{t,C}$. (b) If the current observation does not fit within the local map due to the agent's location, we add zero-padding (gray) to both the observation and the local map. Hence, the size of the local map increases ($H$ changes).

We also store the ratio $q_c$ of the number of frontier cells ($N_{\text{frontier}}$) to the number of known cells ($N_{\text{known}}$) and the distance between each fracture point in the current local map $\mathbf{M}_t^{\text{cur}}$, as further explained in this section. The ratio is used for guiding agents on whether or not to move to other local maps. When $\mathbf{M}_t^{\text{cur}}$ is stored in LTM, it is connected with adjacent map fragments that share the same fracture point in the connectivity graph. In other words, the node of the graph is a model fragment, and a connection denotes that both fragments share a fracture point. In our implementation, we designed a fracture border to prevent unnecessary overlaps and over-fragmentation. The fracture border extends from a fracture point to the left and right based on the agent's head direction until the border reaches either the frontier or an occupied cell. When the agent crosses this border, the recall process is triggered. This mechanism ensures that the agent does not need to be exactly on a fracture point to recall existing maps, thus enhancing flexibility and preventing over-fragmentation. The fracture points are themselves used as subgoals to switch to adjacent map fragments mentioned in Section 3.6. Note that an agent does not need to fragment an environment if it is a single open space, such as a large rectangular or circular arena.

**Recall** Each local map records the fracture points. At these points, there are overlaps with other map fragments. When the agent moves to a fracture point in the current local map, the corresponding local map is recalled from LTM and the current one is stored in LTM.

### 3.5 Surprisal

The surprisal serves as a criterion for fragmentation, which can be any uncertainty estimate of the future, such as negative confidence or future prediction error. We employ the local predictive map for measuring surprisal. The scalar surprisal signal $s_t = 1 - c_t$ is generated using the local map in STM and the current observation, where $c_t$ quantifies the average similarity of the visible part of the observation to the local predictive map $\mathbf{M}_{t-1}^{\text{cur}}$ before update:

$$c_t = \frac{\mathbf{M}_{t-1,C}^{\text{cur}} \cdot o'_{t,C}}{||o'_{t,C}||_1}. \tag{2}$$

The agent is assumed to maintain a running estimate of the mean $\mu_t$ and standard deviation $\sigma_t$ of past surprisals, stored as part of the current map.

### 3.6 Subgoal

Subgoals are decided by using either the current local map in STM or the connectivity graph in LTM. The former enlarges the current local map while the latter helps find the next local map to explore. An agent explores the local region in the environment unless the current surprisal is too low (*e.g.*, *z*-score is smaller than $-1$) and there is a less explored local map nearby.

Subgoals made with the current local map are based on frontier-based subgoals (Yamauchi, 1997) for exploring the local region. Each cell in the region is categorized as known and unknown based on whether it was previously observed or not, and occupied and unoccupied (empty) based on its occupancy. In the current local map, we first find all frontiers which are unknown cells adjacent to the known unoccupied cells. A group of consecutive frontiers is called a 'frontier-edge' and Yamauchi (1997) uses the nearest centroid of the frontier-edge as a subgoal. Unlike standard SLAM methods that employ the entire map, our map in STM only covers a subregion of the environment. After fragmentation, the region where the agent came from has several frontiers (border of two local models) forming a frontier-edge. It leads the agent to go back to the previous area and recall the corresponding map fragment. This would lead to the agent moving between two map fragments for a long time. Therefore, we prioritize the frontier-edge that is not located spatially behind the agent. The subgoal is sampled with the following weight $w_i$ for each frontier-edge $\mathcal{F}_i$:

$$w_i = \frac{|\mathcal{F}_i| \cdot \mathbb{1}(\mathcal{F}_i \text{ is not located spatially behind the agent})}{d_i}, \tag{3}$$

where $d_i$ is the distance between the current position and the centroid of $\mathcal{F}_i$ and $\mathbb{1}(\cdot)$ is the indicator function that is 1 if the condition is true otherwise 0.

Once the agent finishes mapping the local region, it should move to different subregions. However, subgoals from the current local map can misguide the agent to the already explored region since the agent does not have information beyond the map. Hence, we employ the connectivity graph of local maps stored in LTM. We leverage the discovery ratio (the ratio of the number of frontier cells to the number of known cells) $q$ mentioned above to find the most desirable subregions to explore. We also utilize the Manhattan distance between the current agent location and the fracture point between the current ($c$-th) local map and the connected $i$-th local map, $d_{i,c}$ where $d_{c,c} = 0$ and $d_{j,c} = \infty$ if the $j$-th local map is disconnected to the current map. Then, the desirable local map is selected as

$$g = \arg\max_i \frac{q_i}{d_{i,c} + \epsilon}, \tag{4}$$

where $\epsilon$ denotes the preference of staying in the current local map; a smaller value encourages staying in the current local map. If $g$ is not equal to $c$, the fracture point between the current local map and the $g$-th local map is set as the subgoal. Once the agent arrives at the fracture point, the corresponding local map is recalled and the agent recursively checks Eq. 4 until $g$ is the arrived subregion. Note that the distances between fracture points stored in the recalled local map are precomputed since they are fixed.

### 3.7 Planner

The planner takes a subgoal and the current spatial map in STM and finds the shortest path within the map from the current agent location to the subgoal. We use Dijkstra's algorithm for planning a path to the next subgoal. However, the planner can be any path planning method such as the A$^*$ algorithm (Hart et al., 1968) or RRT (LaValle, 1998).

## 4 Procedurally-Generated Environment

We build a procedurally-generated environment for the map-building experiments. Figure 9 and Algorithm 2 in Appendix show the procedure of map generation. We first generate grid-patterned square rooms and randomly connect and merge them. Then, we flip boundary cells (empty or occupied) multiple times for diversity. Formally, given the length of a square $S$, the interval between square rooms, $L$, and the size of the grid, $(N, M)$, we first generate the binary square grid map $\mathcal{M} \in \{0(\text{empty}), 1(\text{occupied})\}^{(N \cdot S + (N+1) \cdot L) \times (M \cdot S + (M+1) \cdot L)}$. Let

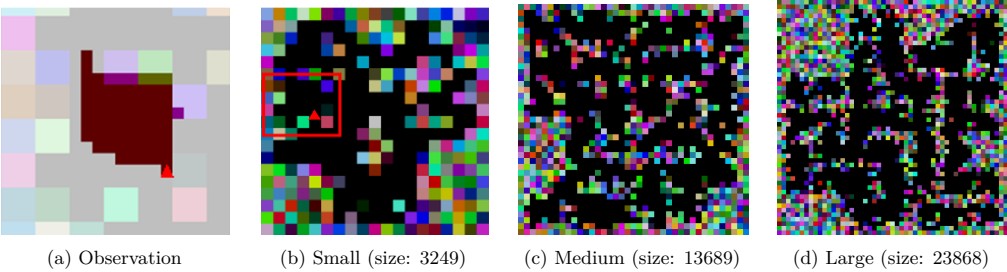

(a) Observation    (b) Small (size: 3249)    (c) Medium (size: 13689)    (d) Large (size: 23868)

Figure 4: Environments. Empty cells (that can be occupied by the agent) are black; walls are randomly colored. (a) Top-down visualization of the agent's local field of view (FOV) (agent: red triangle; shaded region: observation) within an environment (b). The agent has only a locally restricted egocentric view. The right side is occluded by a wall. (b) Top-down view of one environment. The red box marks the region shown in (a). (c), (d) Examples of medium and large environments.

$s_i$ be the $i$-th square in a row-major order in $\mathcal{M}$. For each of the adjacent square pairs, we connect two squares with probability $p_{\text{connect}}$ as a width $w \sim \text{unif}\{1, 2, \ldots, S-1\}$ or merge (a special case of connecting with width $S$) them with probability $p_{\text{merge}}$. Then, we flip all boundaries between occupied and empty cells $K$ times with probability $p_{\text{flip}}$. After flipping the boundaries, there are several isolated (*i.e.*, not connected to other submaps) submaps in $\mathcal{M}$. We only use the submaps where the sizes are greater than a threshold ($3S^2$ in our implementation). After creating maps, we randomly colorize each occupied cell and scale them up by a factor of 3. Note that the proposed environment has much more complex maps compared to Minigrid (Chevalier-Boisvert et al., 2018). Please refer to Appendices D and F.1 for more details.

Figure 4 shows examples of environments and observation. The walls in the environment are randomly colored and are composed of various narrow and wide pathways. For each trial, the agent is randomly placed before it begins to explore the environment. Figure 4a illustrates an example of the agent's view in the small environment shown in Figure 4b. The agent is presented as a red triangle and the observed cells are shaded. The agent has a restricted field of view with occlusion (130°).

## 5 Experiments

In this section, we conduct experiments for FARMap comparing with its baselines on the proposed procedurally generated map environments and robot simulations. We conduct an ablation study, and a sensitivity analysis of hyperparameters in Appendices H and I, respectively. To quantify the difficulty of the proposed environments for the RL exploration algorithm, we measure the performance of RND (Burda et al., 2019) in the environments in Appendix J.

We measure the map coverage, memory usage, and wall-clock time for each environment at each time step as our evaluation criteria and calculate the mean and standard deviation over all runs. The memory usage in each environment is calculated as a ratio of the local map size (memory size, $H \times W$) to the environment size. Note that the local map size is the asymptotically dominant factor in the memory. We compare FARMap with standard frontier-based exploration (Frontier) (Yamauchi, 1997). Please refer to Appendix F for the experimental settings.

### 5.1 Comparison with Grid Cell Remapping

We conduct experiments within simulated environments that replicate existing rat studies (Derdikman et al., 2009; Carpenter et al., 2015) to validate that the fracture points generated by FARMap correspond with the actual remapping locations where activation patterns are changed in these experiments. Since there is no established metric for quantifying remapping patterns of grid cells in neuroscience, we qualitatively compare the fracture points with the remapping locations. Figure 5 illustrates that the fracture points align closely with the actual remapping locations of rats' grid cells observed in the experiments. This alignment can be attributed to our agent's egocentric view and its limited field of view, similar to that of the rats in the

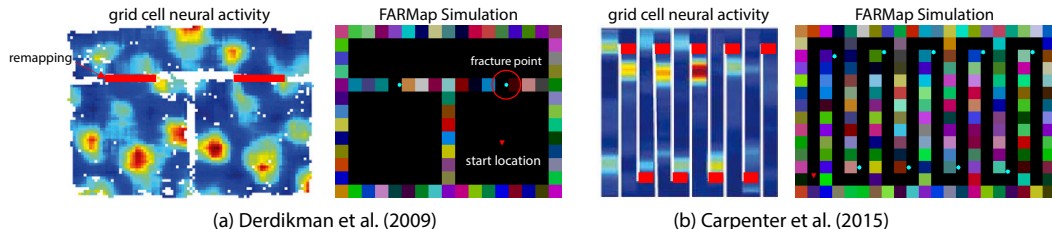

Figure 5: Comparison between remapping locations of grid cells in neuroscience experiments (Derdikman et al., 2009; Carpenter et al., 2015) and fracture points of FARMap in simulation. The red rectangles and emerald circles denote the actual remapping locations and fracture points, and the red triangle is the start location of the simulation. The fracture points are well aligned with the actual remapping locations.

experiment. When the agent passes through a narrow pathway (Figure 5a) or turns a corner (Figure 5b), it encounters new observations that were previously occluded. These new observations significantly increase the surprisal. When the surprisal drops again below a threshold from above, it triggers a fragmentation event, as in Klukas et al. (2021). The difference in surprisal before and after these events is due to the sudden exposure to new, unpredicted information, which is more pronounced after the agent has turned or moved past the occlusion.

## 5.2 FARMap in Procedurally-Generated Environments

We conduct experiments on 1,500 different environments to show that FARMap can explore new environments without prior training. Note that FARMap is not a learning algorithm; rather, it operates similarly to Frontier and other SLAM methods by efficiently exploring new environments without learning. Figure 6 summarizes the performance over the course of exploration on 1,500 environments with three groups based on their sizes; small (size < 5,000), medium (5,000 ≤ size < 15,000), and large (size ≥ 15,000). The lines in the plots are the average of all experiments or a group of experiments and the shaded areas are standard errors of the mean which are not visible due to a large number of trials. FARMap clearly outperforms the baseline on every step, which means that it explores the environment more efficiently. On the other hand, FARMap generally uses a stable amount of memory on average (40 %) over all experiments while Frontier requires much more memory as map coverage increases. The average memory usage of FARMap is almost consistent in any group of environments as the agent explores environments while the usage of Frontier keeps increasing.

Figure 7 and Table 1 analyze memory size and wall-clock-time changes depending on the environment size. 'Random Exploration' denotes an agent moving randomly at every step. Although its runtime is fast, it cannot explore as many areas when the environment becomes larger or more complex. The memory usage of FARMap in each environment is measured by the biggest memory size during exploration since the size is dynamically changed by fragmentation and recall. FARMap clearly outperforms the baseline with a much less wall-clock time while planning. This is because our agent only refers to the subregion of the environment, not using the entire map. Especially in large environments, it is approximately four times faster than the baseline. Moreover, FARMap requires less memory than the baseline, as we mentioned above. The high confidence intervals are caused by aggregating results from multiple high-variance environments (see Appendix G). We also measure the ratios of memory usage and map coverage and of wall-clock time and map coverage in Table 2. The result shows that FARMap has a smaller ratio in all criteria, which means that it requires fewer time and memory resources to explore 1% of an environment.

## 5.3 Dynamic Environment

Inspired by Random Disco Maze (Badia et al., 2020), we build medium-sized 345 dynamic environments where the wall colors change every time step that contributes to an increase in surprisal due to mismatched predictions. Table 3 shows that all methods work well in the environments, and FARMap retains its efficiency compared to Frontier in terms of memory and wall-clock time.

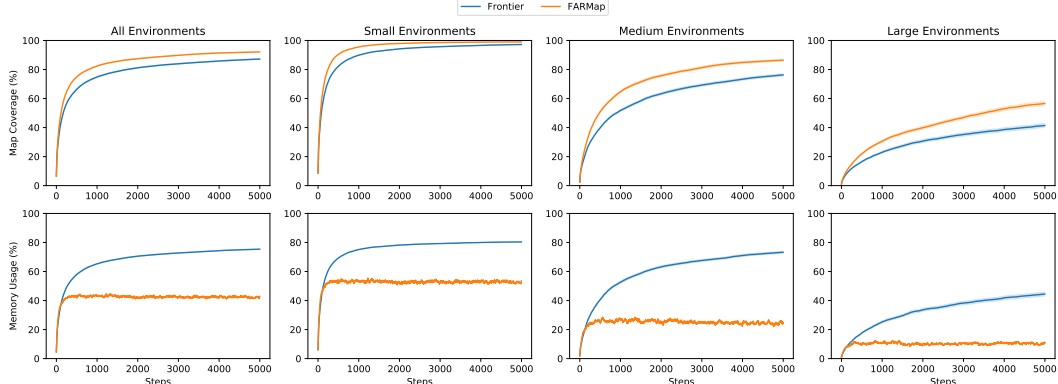

Figure 6: Growth in agent-explored map region as a function of the number of steps in the environment matches the performance of an augmented Frontier-based baseline with less memory use. Mean spatial map coverage performance (top) and mean memory usage (bottom) as a function of the number of steps taken in various sizes of environment sets. FARMap achieves better or comparable exploration than a Frontier-based exploration baseline (Frontier) (Yamauchi, 1997). while using only about half the memory on average. The memory benefit increases in a larger environment.

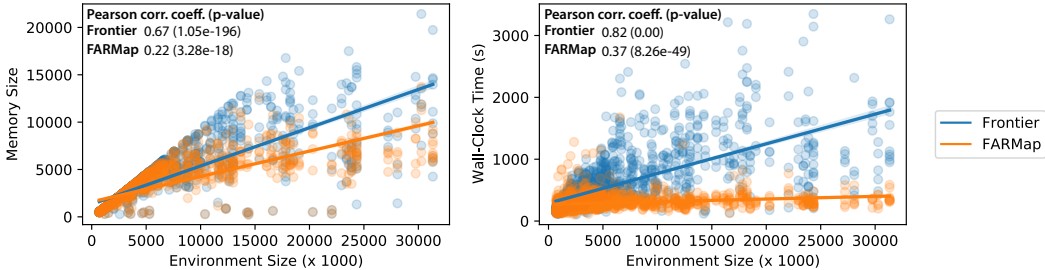

Figure 7: Relative memory and wall-clock time advantage of FARMap to Frontier grow with environment size. Comparison of memory cost (left) and wall-clock time (right) as a function of environment size (circles: experimental results; line: linear regression fit). FARMap requires substantially less memory and is much faster than other methods.

Table 1: Comparison of average map coverage (%), memory use (%), and wall-clock time (*s*) for small, medium, and large environments. The memory usage advantage of FARMap relative to its counterpart grows with environment size. The numbers in parentheses are 95 % confidence intervals generated by bootstrap with one million samples. [†]: Random Exploration does not need memory for exploration.

| Model | Small (size < 5,000) | | | Medium (5,000 ≤ size < 15,000) | | | Large (size ≥ 15,000) | | |
|---|---|---|---|---|---|---|---|---|---|
| | Coverage | Memory | Time | Coverage | Memory | Time | Coverage | Memory | Time |
| Random Exploration | 51.1 (10.3, 96.1) | -[†] | 17.5 (6.0, 52.8) | 30.7 (3.9, 72.5) | - | 30.6 (6.6, 90.6) | 19.9 (2.2, 49.8) | - | 55.8 (8.1, 117.9) |
| Frontier (Yamauchi, 1997) | 97.2 (76.0, 100.0) | 80.4 (61.8, 88.7) | 360.5 (154, 773) | 76.3 (15.6, 99.8) | 73.3 (13.0, 92.3) | 871.9 (290, 2020) | 41.4 (6.1, 84.3) | 44.4 (3.8, 84.3) | 1261.0 (217, 3189) |
| FARMap | **99.0 (96.3, 100.0)** | **79.1 (61.4, 88.0)** | **278.2 (139, 538)** | **86.4 (15.6, 100.0)** | **62.9 (12.5, 90.2)** | **321.4 (191, 528)** | **56.6 (6.1, 97.7)** | **31.4 (3.8, 54.3)** | **352.5 (202, 633)** |

## 5.4 FARMap in Robot Operation Simulation

We simulate FARMap in four continuous environments with TurtleBot3 (Burger) via Robot Operation System (ROS) (Macenski et al., 2022) with Gazebo simulator. ROS is one of the standard libraries for conducting robotic experiments, and it allows for straightforward deployment to real robots at no additional cost. Unlike experiments performed in Section 5.2, the observation here involves a 360-degree first-person view via the default laser scan. We utilize the default global planner in the 'move_base' package. Frontier and FARMap are tested in four continuous 3D environments with a fixed starting location (Figure 8), for

Table 2: Comparison of the ratios of memory usage and map coverage, and of wall-clock time and map coverage. Smaller value denotes the model is more efficient than others. FARMap has the smallest ratios in all comparisons.

| Model | Small | | Medium | | Large | |
|---|---|---|---|---|---|---|
| | Memory / Coverage | Time /Coverage | Memory /Coverage | Time / Coverage | Memory /Coverage | Time / Coverage |
| Frontier | 0.83 | 3.71 | 0.96 | 11.43 | 1.07 | 30.46 |
| FARMap | **0.80** | **3.52** | **0.73** | **3.72** | **0.55** | **6.22** |

Table 3: All methods have stable performance in dynamic environments. We measure average map coverage (%), memory use (%), and wall-clock time (s) for dynamic environments with 95% confidence intervals computed by bootstrap with one million samples.

| Method | Coverage (%) | Memory (%) | Time (s) |
|---|---|---|---|
| Frontier | 95.0 (72.2, 100.0) | 86.6 (64.5, 90.0) | 742.2 (385.6, 1361.7) |
| FARMap | **95.5 (72.5, 100.0)** | **67.9 (37.0, 89.6)** | **386.0 (154.5, 521.7)** |

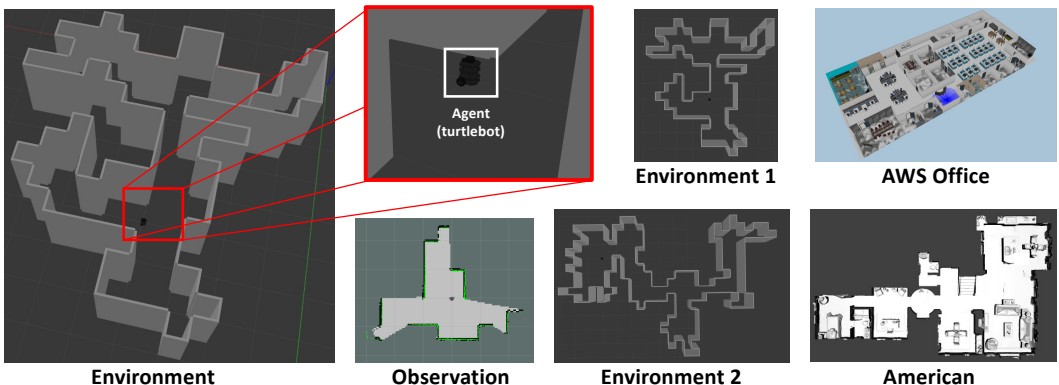

Figure 8: Robot simulation environments. The turtlebot agent moves around with a 360-degree laser scan sensor to map the entire space.

2500 steps using five different random seeds. The laser scan operates at a frequency of 2.5Hz, meaning that the agent updates the local map every 0.4 seconds.

Table 4 presents a comparison between FARMap and Frontier in terms of map coverage and memory usage measurements without any normalization. We do not use wall-clock time for the comparison as it is now related to the agent step. In most environments, FARMap has better exploration performance with less memory. Although FARMap consumes more memory than Frontier in the AWS Office, its memory-to-coverage ratio is better than Frontier's (1.16 compared to 1.26, respectively).

## 5.5 FARMap with Neural SLAM

We conducted experiments on both FARMap and Frontier integrated with the pre-trained Neural SLAM (Chaplot et al., 2020) obtained from the official repository for the Gibson (Shen et al., 2021) exploration task with the Habitat simulator (Szot et al., 2021). We use 'American' used in Section 5.4 as an example. For the fair comparison with FARMap and Neural SLAM, we replaced the global policy in FARMap or Frontier to establish the 'long-term goal', following Chaplot et al. (2020). This essentially means that we employ a Neural SLAM module to convert RGB observations to a 2D map and a Local Policy to generate discrete actions based on the given global goal. Table 5 demonstrates that Neural SLAM, when substituting FARMap for global policy, attains superior exploration performance. In contrast, incorporating Frontier led to a decrement in performance. These experimental outcomes also hint at the potential advantages of applying our fragmentation-and-recall concept to exploration methods that leverage maps.

Table 4: FARMap has better performance with less memory and time in 3D robot simulation environment 1 while it has similar performance with more time in environment 2. The number in parenthesis denotes a 95% confidence interval.

| Model | Environment 1 | | Environment 2 | | AWS Office (Erdogan, 2019) | | American (Shen et al., 2021) | |
|---|---|---|---|---|---|---|---|---|
| | Coverage (k) | Memory (k) | Coverage (k) | Memory (k) | Coverage (k) | Memory (k) | Coverage (k) | Memory (k) |
| Frontier | 7.0 (± 1.4) | 20.5 (± 1.0) | **8.3 (± 0.6)** | 32.8 (± 34.4) | 38.2 (± 30.0) | **48.1 (± 20.8)** | 13.8 (± 3.1) | 11.0 ( ± 2.1) |
| FARMap | **7.7 (± 1.0)** | **20.1 (± 2.4)** | 8.3 (± 0.1) | **23.0 (± 8.6)** | **57.0 (± 4.7)** | 66.0 (± 14.3) | **15.8 (± 4.2)** | **10.6 (± 3.7)** |

Table 5: Comparison of Neural SLAM and its adaptations with Frontier and FARMap on the Gibson American environment.

| Model | % Cov. | Cov. (m$^2$) |
|---|---|---|
| Neural SLAM (Chaplot et al., 2020) | 0.818 | 64.795 |
| Neural SLAM w/o global policy + Frontier | 0.733 | 58.103 |
| Neural SLAM w/o global policy + FARMap | **0.833** | **66.012** |

# 6    Discussion

We have proposed a new framework for exploration based on local models and fragmentation, inspired by how natural agents explore space efficiently through grid cells' remapping. Our framework dynamically *fragments* the exploration space based on the current surprisal in real time and stores the current model fragment in long-term memory (LTM). Stored fragments are *recalled* when the agent returns to the state where the fragmentation happened so that the agent can reuse the local information. Accordingly, the agent can refer to *longer-term* local information. This method shows potential for broad application in tasks involving streaming observations or data that are recurrent or reused (Hwang et al., 2023). Specifically, we have applied this to the setting of spatial exploration. The surprisal is generated by short-term memory (STM) using a local map in FARMap. FARMap closely replicates the fragmentation behavior observed in animal studies. This alignment with biological systems underscores the potential of our framework for capturing essential aspects of natural exploration processes.

FARMap outperforms the baseline method (Yamauchi, 1997) in terms of reduced wall-clock time, memory requirements, and action count while enhancing map-building performance in both static and dynamic discrete environments as well as in continual robot simulations. Considering that Yamauchi (1997) is still a core algorithm in the recent state-of-the-art SLAM methods (Bonetto et al., 2022; Hess et al., 2016; Placed et al., 2022) (*e.g.*, Figure 12), FARMap is complementary in that it addresses a key inefficiency of global map-based approaches, which is that they can be memory-intensive and computationally demanding.

Our paper aims to be a proof-of-concept for fragmentation and recall in spatial map-building inspired by biological principles using frontier-based exploration and Neural SLAM (Chaplot et al., 2020). We believe that this concept can be applicable to other exploration paradigms and various applications (see Appendix E). This concept can make large-scale exploration, which typically requires a huge memory size and long-ranged memory span, significantly more efficient.

**Broader Impact Statement**

Our main focus of this work is to connect neuroscience and spatial exploration so that two different research communities interact more actively with each other. Our method can be exploited for military purposes like other spatial exploration methods or SLAM.

**Acknowledgement**

We appreciate Jim Neidhoefer helping implement the robot simulation experiment. This research was supported by ARO MURI W911NF2310277 and Ila Fiete is supported by the Office of Naval Research, the Howard Hughes Medical Institute (HHMI), and NIH (NIMH-MH129046).

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

## Appendix

## A   Bridging between Neuroscience and Machine Learning

Our contributions hold significant relevance for the machine learning (ML) community, beyond robotic SLAM. They also help to connect back to neuroscience, by generating functional hypotheses that can be tested in the brain.

Our interdisciplinary approach, which leverages biological principles towards making broader AI advancements, has the potential to develop more robust and efficient AI systems and inspire novel ML algorithms. The Fragmentation-and-Recall framework is not only effective with traditional SLAM (Yamauchi, 1997) but also improves the exploration performance of SLAM based on neural networks (Chaplot et al., 2020). This shows the potential of FARMap when it combines with a neural network-based spatial exploration approach.

Further, FARMap is designed as a fundamental algorithm that should be relevant beyond spatial tasks like robotic navigation. For instance, it can be leveraged for any model-based learning, from motor control to reinforcement learning. The agent builds local models (where "local" may be in terms of spatial location for a navigating agent or in state space for motor control, for instance). When the local model fails to predict the next states well, the agent may select a different local model or build a new model. Thus, the concept of

surprisal-based clustering and memory-efficient mapping can be extended to other areas of machine learning, such as spatial exploration, memory optimization, reinforcement learning, and autonomous decision-making, and goal-directed behavior.

Conversely, in neuroscience, where the phenomenon of spatial fragmentation was observed, its implications for function have not yet been studied or appreciated. FARMap supplies hypotheses for the functional roles of the observed map fragmentation, motivating experiments to test these hypotheses. In addition, FARMap provides a unifying algorithm for episodic memory, spatial navigation, and perhaps also segmented motor control. It predicts that a universal "surprisal" signal might exist in the brain and play a central role in signaling and inducing model or map fragmentations across behavioral domains. Thus, FARMap motivates new experiments in neuroscience: to search for signals that trigger the building of entirely new models or maps, across domains of spatial mapping, cognitive modeling, and motor control.

## B  Additional Related Works

### B.1  Graph-Based SLAM

Graph-based SLAM (Grisetti et al., 2010; Yang et al., 2021; Kulkarni et al., 2022) constructs a topological graph for efficient exploration by reducing the dimensionality of the planning problem. Once this graph is established, a planner utilizes it to navigate toward subgoals. GBPlanner (Yang et al., 2021; Kulkarni et al., 2022) creates a random graph in the local region and uses it for path planning. This reduces computational cost for local path planning by reusing sparse graph nodes although it still uses a frontier. In contrast, FARMap aims for efficient exploration in terms of memory, time, and the number of steps by dividing the environment (*i.e.*, fragmentation) and the topological graph is used for moving one subregion to another. We believe that there is a potential synergy between graph-based SLAM and FARMap. Such synergy can be achieved by substituting frontier-based exploration with a graph-based approach, pairing global fragmentation from FARMap with relatively local planning from graph-based methods.

### B.2  Reinforcement Learning in Neuroscience

Many animal experiments involving goal-directed learning combine rewards (or punishments) with tasks, such as evidence accumulation (Mochizuki-Freeman et al., 2022; Nieh et al., 2021; Lee et al., 2022; Pinto et al., 2019) and simple visual cues (Vorhees & Williams, 2006). This has led computational neuroscientists to use reinforcement learning (RL) to model brain functions, resulting in models with good explanatory power for both neural and behavioral data (Niv, 2009; Pedamonti et al., 2023; Recanatesi et al., 2021; Vorhees & Williams, 2006). However, spatial learning (Tolman, 1948) served as a powerful rebuttal to reward-based learning and the behaviorist school in Psychology: Animals typically acquire and represent spatial information even in the absence of spatially-conditioned rewards. Grid and place cells form spatial representations, and grid cell remapping is observed in free-moving animals with no operant or reinforcement-based training. Thus, goal-directed learning is not required for spatial representations and remapping. However, spatial representations and remapping could very well play an important role in spatial reinforcement learning. Integrating the fragmentation-and-recall framework into goal-conditioned RL is an intriguing direction for future research, beyond the scope of the present paper.

### B.3  Memory-Based Reinforcement Learning

Although the reinforcement learning (RL) algorithm is beyond the scope of this paper, FARMap is similar to memory-based RL in the sense that it uses memories. Hung et al. (2019) combine LSTM (Hochreiter & Schmidhuber, 1997) with external memory, along with an encoder and decoder for the memory. Ritter et al. (2018a;b) use DND (Pritzel et al., 2017) to store the states of LSTM with its inputs and retrieve old states to update the state of LTM in meta-reinforcement learning tasks. Similarly, Fortunato et al. (2019) use working memory and an episodic memory structure but employ an output of the episodic memory as an input for the working memory. On the other hand, Lampinen et al. (2021) utilize a hierarchical LTM with chunks and attention for long-term recall inspired by Transformers (Vaswani et al., 2017) however, their chunks are formed periodically rather than based on content and are not used as intrinsic options for

exploration. Our spatial map-building framework is similar to memory-based RL methods in terms of having two memory architectures inspired by the brain. However, FARMap fragments an environment (or space) in an online manner and recalls stored memories inspired by grid cells, while memory-based RL stores previous states. Moreover, we use the connectivity graph of STMs to find the next subgoal for efficient map building. We would like to emphasize that FARMap is not a reinforcement learning method. On the other hand, we believe that our proposed concept, *fragmentation-and-recall* can be applicable to memory-based reinforcement learning by reducing search space in the memory.

## C  LTM Retrieval Overhead

FARMap needs to consider the retrieval time of LTM since it is not located in the main memory. If the memory (RAM) is larger than the environment so that we can even use LTM on RAM, retrieval time is not a concern, and FARMap is useful in boosting speed, although it might use more memory. In our original scenarios, LTM is an external memory (non-volatile memory). Usually, SSD's speed (including bandwidth and read/write) is around 300-600 MB/s while RAM (DDR4) operates at 5-25 GBps. In this case, SSD read/write can be a bottleneck. However, the flash memory speed is around 5 GBps, and the retrieval time for the map will be negligible compared to the planning time. It is generally not recommended to use a hard disk drive (HDD), whose data transfer rate is around 100 MB/s.

## D  Discussion about the Proposed Environment

### D.1  Wall Color and Exploration

When the environment is static, the color of the wall does not affect FARMap; the model can still detect surprisal events effectively even if the wall color remains constant. On the other hand, in dynamic environments, where the color of walls changes at each time step (as discussed in Section 5.3), the color changes contribute to an increase in surprisal due to mismatched predictions. Despite this increased surprisal, FARMap remains effective in detecting and managing fragmentation events, demonstrating its robustness in both static and dynamic scenarios.

### D.2  Comparison with MiniGrid

MiniGrid (Chevalier-Boisvert et al., 2018) is specifically designed for Reinforcement Learning (RL) experiments and involves tasks that require an agent to understand and navigate environments by avoiding hazards, picking up colored keys, opening doors, and recognizing goals. This setup necessitates an RL policy, whereas FARMap is not based on reinforcement learning algorithms. Additionally, the MiniGrid environment is composed of simple rectangular rooms and corridors, which do not provide the level of complexity we require for evaluating our method. To thoroughly test FARMap's capabilities, we use more complex maze environments that better challenge the agent's ability to manage fragmentation and recall in a variety of spatial configurations.

## E  Potential Applications

In this section, we introduce several potential applications where FARMap can be helpful by reducing memory and time costs.

### E.1  Mars Exploration

Mars exploration rovers such as Opportunity and Curiosity have limited resources. For example, the Curiosity rover has 256 MB of RAM and 2GB of flash memory[2]. However, the mission range on Mars may be much larger than the RAM. Therefore, efficient mapping is required and we believe that FARMap could be helpful in Mars exploration.

---

[2]https://mars.nasa.gov/msl/spacecraft/rover/brains/

Table 6: The statistics of the size of environments in the dataset.

| Statistics | All | Small | Medium | Large |
|---|---|---|---|---|
| The number of environments | 1500 | 1015 | 345 | 140 |
| Average size | 5697.8 | 2466.7 | 8532.4 | 22138.7 |
| Standard deviation of size | 6265.8 | 1253.7 | 2828.5 | 4872.9 |

### E.2  2D/3D Mapping with LiDAR

As mentioned in Section 5.4, FARMap is capable of utilizing observations from LiDAR for map-building in continuous environments. The resolution of the sensor can be set to a cell unit. Considering the properties of the Robot Operating System (ROS) (Macenski et al., 2022), we believe that FARMap can be easily deployed to a real robot. Additionally, it is feasible to extend it to 3D by using 3D voxel mapping instead of 2D pixel mapping. This approach can prove beneficial in large-scale environments such as buildings, airports, and houses.

## F  Experimental Details

Our models are implemented on PyTorch and the experiments are conducted on an Intel(R) Xeon(R) CPU E5-2650 v4 @ 2.20GHz for spatial exploration experiments and on an NVIDIA Titan V for RND and Neural SLAM.

### F.1  FARMap Environment Generation

To generate the environment, we run map generation (Algorithm 2) 200 times and then use the 300 largest-sized maps. All maps are scaled up by a factor of 3 after colorization for the task. On every trial, we sample $S$ and $N$ from $\{3, 4, 5, 6, 7\}$ and set $M = N$. $K, L \in \mathbb{N}$ are sampled from $[0, 10]$ and $[1, 3]$, respectively. We set $p_{\text{connect}}$, $p_{\text{merge}}$ and $p_{\text{flip}}$ to 0.25, 0.25, and 0.05, respectively. Figure 9 illustrates the procedure of environment generation described in Algorithm 2. Table 6 shows the statistics of the size of the generated environments.

### F.2  FARMap

We run the agent on 1,500 different environments: 300 different maps with five random seeds and the starting position and the color of the map are changed on each random seed. We set $\gamma$, $\rho$, and $\epsilon$ to 0.9, 2, and 5, respectively. The observation size $(h, w)$ is (15,15). If the frontier-based exploring agent is surrounded by a large frontier-edge in an open space, the centroid of the frontier can fall into the interior of the explored space, leading to no new discovery. This causes the agent to become stuck. We improve the agent by selecting the nearest unoccupied cell from the nearest frontier state to the centroid.

### F.3  RND

We train RND (Burda et al., 2019) for 1 million steps without extrinsic reward for each environment. RND is based on recurrent PPO (Schulman et al., 2017). Table 7 shows the architecture of RND used for the experiments. The learning rate is 0.0001, the reward discount factor is 0.99 and the number of epochs is 4. For other parameters, we use the same values mentioned in PPO and RND: we set the GAE parameter $\lambda$ as 0.95, value loss coefficient as 1.0, entropy loss coefficient as 0.001, and clip ratio ($\epsilon$ in Eq. 7 in Schulman et al. (2017)) as 0.1.

## G  Wide Confidence Intervals

In Table 1, 95% confidence intervals for each measurement are generated by bootstrapping with one million samples. The confidence intervals are very wide because our metrics (map coverage, memory usage, and

---

**Algorithm 2** Spatial Exploration Environment Generation

---

**Require:** $N, M, L, S, K, p_{\text{connect}}, p_{\text{merge}}, p_{\text{flip}}$
**Ensure:** A set of maps, $\{\mathcal{M}\}$.

1: **procedure** MapGeneration
2:    Initialize $\mathcal{M} \in \{0,1\}^{(N \cdot S + (N+1) \cdot L) \times (M \cdot S + (M+1) \cdot L)}$, $(N, M)$ grid with interval $L$ and each square sized $(S, S)$.                              ▷ Figure 9 (1).
3:    **for** $(s_i, s_j) \in \{(s_i, s_j) | s_i \text{ and } s_j \text{ are adjacent}, i \leq j\}$ **do**         ▷ Get adjacent grid square pairs.
4:        $x \sim \mathcal{B}(1, p_{\text{connect}})$                    ▷ Connect adjacent squares with probability $p_{\text{connect}}$.
5:        **if** $x = 1$ **then**
6:            $w \sim \text{unif}\{1, \ldots, S - 1\}$
7:            Connect $s_i$ and $s_j$ with width $w$.                                 ▷ Figure 9 (2).
8:        **end if**
9:        $x \sim \mathcal{B}(1, p_{\text{merge}})$                       ▷ Merge adjacent squares with probability $p_{\text{merge}}$.
10:        **if** $x = 1$ **then**
11:            Merge $s_i$ and $s_j$ by removing the interval.                        ▷ Figure 9 (3).
12:        **end if**
13:    **end for**
14:    **for** $k \leftarrow 1$ **to** $K$ **do**
15:        **for** $c \in \{c | c \in \mathcal{M}, \exists_{c'} \, c \text{ xor } c' = 1, c' \in \text{Adj}(c)\}$ **do**         ▷ Get boundary cells in the map.
16:            $x \sim \mathcal{B}(1, p_{\text{flip}})$                               ▷ Flip the cell with probability $p_{\text{flip}}$.
17:            $c = c \text{ xor } x$                                          ▷ Figure 9 (4)-(6).
18:        **end for**
19:    **end for**
20:    Divide $\mathcal{M}$ into a set of isolated maps $\{\mathbf{m}_i\}$                      ▷ Figure 9 (7).
21:    Filter out a map in $\{\mathbf{m}_i\}$, where the size is smaller than $3S^2$.
22:    Randomly colorize the occupied cell in each map.                        ▷ Figure 9 (8).
23:    Scale up each map in $\{\mathbf{m}_i\}$ by factor of 3.
24: **end procedure**

---

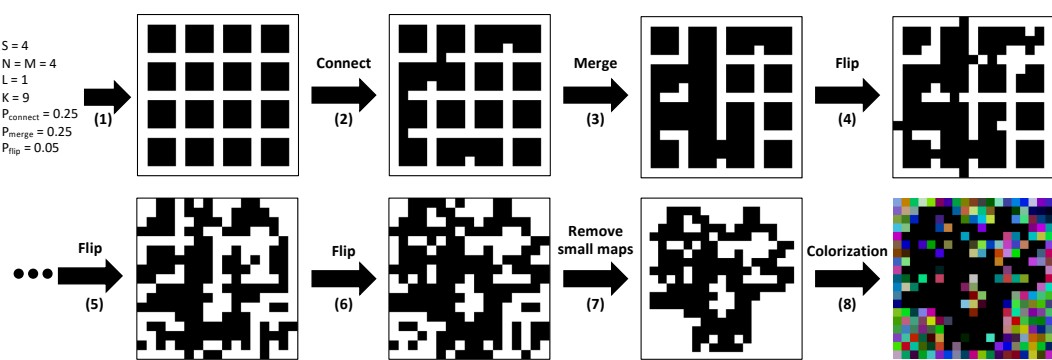

Figure 9: Procedure of map generation. (1) We first set square grid where black and white denote empty and occupied, respectively. (2) We randomly connect and (3) merge adjacent grid. (4)-(6) We also randomly flip the boundaries of empty and occupied cells recursively. (7) Then, we remove small isolated subregions and (8) randomly colorize occupied cells. Finally, we increase the size of the map.

wall-clock time) depend on the size and the complexity of the environment, and each method is evaluated on many varied environments as shown in Table 6 and the code repository.

We also present results with much smaller groups in Figure 10. We first sort the environments based on their sizes, and then we partition the environments into 150 groups, each of size 10, and calculate the average with bootstrapping to get a 95 % confidence interval for each group. The 95% confidence intervals measured by bootstrapping are also smaller than the reported range in Table 1. In particular, FARMap has a relatively

Table 7: The architecture of RND agent. The networks are divided into the policy module and RND module.

| Policy module | RND module |
|---|---|
| Conv2d (8×8, 16) | Conv2d (8×8, 32) |
| Conv2d (4×4, 32) | Conv2d (4×4, 64) |
| FC (3200×512) | Conv2d (3×3, 64) |
| LSTM (512, 512) | FC (3136×512) |
| FC (512×5) × 2 | FC (512×512) |
| FC (512×1) × 2 | FC (512×512) |

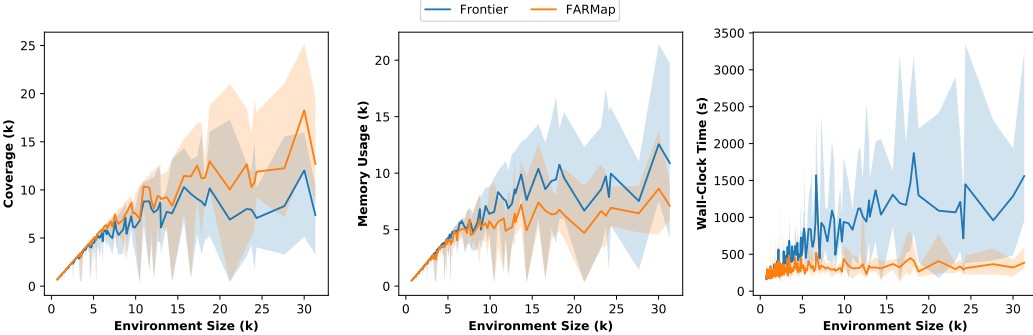

Figure 10: Map coverage, memory usage, and wall-clock time advantage of FARMap to Frontier grow with environment size. Comparison of these metrics as a function of environment size. The mean (line) and 95% confidence interval (shade) are calculated by bootstrapping with one million samples each from 150 groups (10 environments each) ordered by size.

steady wall-clock time across the entire environments while Frontier requires more time with high variance depending on the environments. Although the gaps between FARMap and Frontier in all metrics are small in small environments, they become larger as the environment size grows. In other words, FARMap is better than Frontier in all environments in terms of map coverage, memory usage, and wall-clock time.

## H   Ablation Study

Table 8 illustrates the ablation study of components of FARMap. Each component contributes to improving the performance while it increases memory and time which are negligible compared to the baseline performance (44.4, 1261.0, respectively). We also evaluated FARMap, random fragmentation, and uniform fragmentation methods on top of FARMap. This is to demonstrate that surprisal produces effective fragmentations that maintain the exploration performance with low memory and fast wall-clock time. Random and Uniform models only change the fragmentation criteria and other parts (*e.g.*, LTM, subgoal selection, and planning) remain the same. Table 9 shows that there is a trade-off between frequency of fragmentation and memory usage and wall-clock time. FARMap achieves better exploration performance than random and uniform fragmentation models. Table 10 shows Frontier and FARMap with RRT planner (LaValle, 1998) in large environment. Although the performance gaps between the two models are reduced, FARMap still outperforms Frontier.

Table 8: Ablation study for each component in large environments. We chose the best-performed fragmentation threshold without the $z$-score variant in $[0.1, 0.9]$.

| $z$-score | LTM subgoal | Coverage | Memory | Time |
|:---:|:---:|:---:|:---:|:---:|
| - | - | 51.7 (22.1, 90.5) | **20.8 (10.0, 49.6)** | **177.3 (121.7, 254.8)** |
| ✓ | - | 52.8 (21.9, 85.6) | 30.4 (15.3, 57.7) | 302.3 (167.9, 519.2) |
| ✓ | ✓ | **56.6 (6.1, 97.7)** | 31.4 (3.8, 54.3) | 352.5 (202.0, 633.0) |

Table 9: Comparison of vanilla FARMap and FARMap with random, and uniform fragmentation in Large Environments. The random model decides to fragment with probability on every time step and the uniform model makes a fragmentation on every Interval step ($L$).

| Model | Coverage | Memory | Time |
|:---:|:---:|:---:|:---:|
| FARMap | **56.6 (6.1, 97.7)** | 31.4 (3.8, 54.3) | 352.5 (202, 633) |
| Random ($P = 0.1$) | 45.2 (15.4, 82.3) | 7.8 (4.4, 13.2) | **111.3 (70.9, 141.0)** |
| Random ($P = 0.05$) | 47.5 (18.0, 87.4) | 12.1 (6.7, 20.0) | 136.5 (179.8) |
| Random ($P = 0.01$) | 49.0 (18.6, 87.6) | 24.5 (12.9, 43.7) | 290.7 (148.1, 499.6) |
| Random ($P = 0.005$) | 49.1 (20.5, 89.6) | 30.5 (16.1, 60.0) | 378.1 (201.7, 637.7) |
| Random ($P = 0.001$) | 54.1 (23.5, 92.4) | 46.1 (20.9, 81.4) | 683.4 (292.3, 1484.4) |
| Uniform ($L = 25$) | 49.1 (15.8, 82.5) | **7.5 (4.7, 11.8)** | 110.8 (84.6, 143.7) |
| Uniform ($L = 50$) | 48.8 (17.3, 89.5) | 12.6 (6.8, 20.9) | 147.3 (112.9, 200.8) |
| Uniform ($L = 100$) | 48.3 (18.7, 88.8) | 19.3 (10.7, 32.1) | 216.5 (150.3, 330.2) |
| Uniform ($L = 200$) | 48.8 (22.0, 90.0) | 27.5 (14.0, 45.5) | 322.2 (209.1, 612.0) |
| Uniform ($L = 500$) | 52.2 (22.0, 90.0) | 38.2 (16.5, 82.5) | 484.2 (292.9, 840.6) |
| Uniform ($L = 1000$) | 53.4 (22.1, 91.9) | 46.5 (23.4, 87.9) | 712.3 (380.6, 1425.0) |

Table 10: Comparison of average map coverage (%), memory use (%), and wall-clock time ($s$) in large environments. Both Frontier and FARMap use RRT (LaValle, 1998) planner.

| Model | Coverage | Memory | Time |
|:---:|:---:|:---:|:---:|
| Frontier (Yamauchi, 1997) | 46.9 (20.7, 94.0) | 49.7 (22.6, 90.3) | 880.9 (395.6, 1673.1) |
| FARMap | **50.9 (17.3, 91.3)** | **30.4 (13.8, 55.7)** | **318.1 (174.6, 500.6)** |

Table 11: Sensitivity analysis about fragmentation threshold, $\rho$ in FARMap. The numbers in parentheses are the standard deviation.

| $\rho$ | Small (size < 5,000) | | | Medium (5,000 ≤ size < 15,000) | | | Large (size ≥ 15,000) | | |
|:---:|:---:|:---:|:---:|:---:|:---:|:---:|:---:|:---:|:---:|
| | Coverage | Memory | Time | Coverage | Memory | Time | Coverage | Memory | Time |
| 1.0 | **99.1** | **71.5** | **117.9** | 87.1 | **39.6** | **146.2** | **60.9** | **17.9** | **148.4** |
| 1.5 | 99.1 | 75.7 | 158.0 | 87.6 | 50.2 | 180.1 | 59.7 | 23.3 | 188.9 |
| 2.0 (ours) | 99.0 | 79.1 | 278.2 | 86.4 | 62.9 | 321.4 | 56.6 | 31.4 | 352.5 |
| 2.5 | 98.8 | 80.7 | 207.1 | 89.0 | 79.7 | 557.3 | 58.4 | 56.9 | 770.5 |
| 3.0 | 98.8 | 81.5 | 296.1 | **91.0** | 85.0 | 698.2 | 60.9 | 67.9 | 1068.0 |

# I  Sensitivity Analysis for Hyperparameters in FARMap

We test FARMap with various hyperparameters; fragmentation threshold ($\rho$), decaying factor ($\gamma$), and $\epsilon$. All experiments are conducted in the same environments. While comparing one hyperparameter, we fix the

Table 12: Sensitivity analysis about decaying factor, $\gamma$ in Eq. 1. The numbers in parentheses are the standard deviation.

| $\gamma$ | Small (size < 5,000) | | | Medium (5,000 ≤ size < 15,000) | | | Large (size ≥ 15,000) | | |
|---|---|---|---|---|---|---|---|---|---|
| | Coverage | Memory | Time | Coverage | Memory | Time | Coverage | Memory | Time |
| 0.8 | 98.8 | 79.4 | 210.5 | 85.3 | 64.5 | **304.0** | 55.6 | 32.8 | 304.8 |
| 0.9 (ours) | 99.0 | 79.1 | 278.2 | 86.4 | 62.9 | 321.4 | 56.6 | **31.4** | 352.5 |
| 0.95 | **99.1** | **79.0** | **178.3** | 87.3 | **61.3** | 507.5 | 59.2 | 31.9 | **284.7** |
| 0.99 | 99.1 | 80.8 | 262.3 | **89.3** | 76.5 | 453.8 | **60.4** | 46.7 | 541.5 |

Table 13: Sensitivity analysis about $\epsilon$ in Eq. 4. The numbers in parentheses are the standard deviation.

| $\epsilon$ | Small (size < 5,000) | | | Medium (5,000 ≤ size < 15,000) | | | Large (size ≥ 15,000) | | |
|---|---|---|---|---|---|---|---|---|---|
| | Coverage | Memory | Time | Coverage | Memory | Time | Coverage | Memory | Time |
| 1 | 99.0 | 79.1 | 198.0 | **86.8** | 63.0 | 275.3 | 56.6 | 31.5 | 294.8 |
| 3 | **99.0** | **79.1** | 198.1 | 86.7 | 63.0 | **271.3** | 56.5 | 31.5 | **294.5** |
| 5 (ours) | 99.0 | 79.1 | 278.2 | 86.4 | **62.9** | 321.4 | **56.6** | 31.4 | 352.5 |
| 10 | 99.0 | 79.1 | **197.1** | 86.6 | 62.9 | 272.5 | 56.3 | 31.4 | 294.9 |
| 15 | 99.0 | 79.1 | 198.5 | 86.6 | 63.0 | 288.7 | 55.9 | **31.1** | 295.7 |

Table 14: Average map coverage (%), memory use (%), and wall-clock time ($s$) of RND in small, medium, large, and dynamic environments. The numbers in parentheses are 95 % confidence intervals generated by bootstrapping with one million samples across various environments. The memory usage is calculated by the ratio between the number of parameters (7.7M) and each environment size.

| Environment | Coverage (%) | Memory (%) | Time (s) |
|---|---|---|---|
| Small | 77.0 (31.3, 100.0) | 421.7k (157.5k, 1012.5k) | 31.6 (23.9, 40.4) |
| Medium | 37.1 (11.0, 77.0) | 99.7k (53.7k, 151.9k) | 31.2 (23.7, 39.6) |
| Large | 14.9 (3.4, 33.5) | 36.2k (24.4k, 49.7k) | 30.9 (25.0, 39.6) |
| Dynamic | 37.2 (23.9, 35.6) | 99.7k (53.7k, 151.9k) | 29.1 (10.5, 75.8) |

remaining parameters as $\rho = 2.0, \gamma = 0.9, \epsilon = 5$. Table 11 presents the performance of FARMap with different fragmentation thresholds, $\rho$. The smaller value makes it more prone to fragment the space, which means it can use less memory but overly fragment the space. On the other hand, a bigger threshold uses more memory with less fragmentation. Hence, we choose 2 as the threshold value (95% confidence interval if the distribution follows a Gaussian). On the other hand, FARMap is robust to the decaying factor and $\epsilon$ as shown in Tables 12 and 13, respectively.

## J    Reinforcement Learning Method in the Proposed Environments

We run RND (Burda et al., 2019) based on PPO-LSTM (Schulman et al., 2017) to give an example of reinforcement learning exploration method in the proposed procedurally-generated environments. Table 14 shows the performance of RND in static and dynamic environments. To quantify RND's memory usage based on this measurement, we divided the number of parameters (7.7M) by the environment size. Note that it is difficult to compare with FARMap or Frontier directly since the RL agent is trained on each environment before testing it while FARMap and Frontier have no training. However, in both sets of environments, RND has much lower coverage than FARMap but it is much faster since it does not need to update the local map and planning. We also demonstrate the average map coverage across the number of steps in Figure 11.

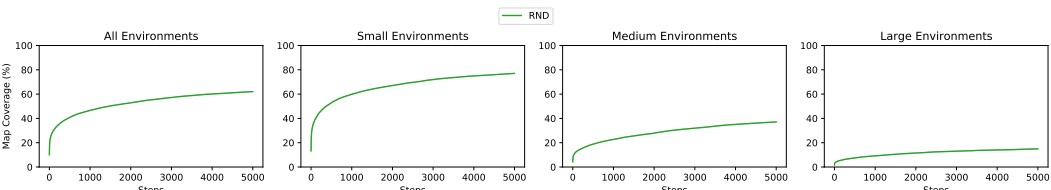

Figure 11: Growth in agent-explored map region as a function of the number of steps from the first step in the environment matches the performance of RND. Mean spatial map coverage performance as a function of the number of steps taken in various sizes of environment sets. The shade denotes the standard error.

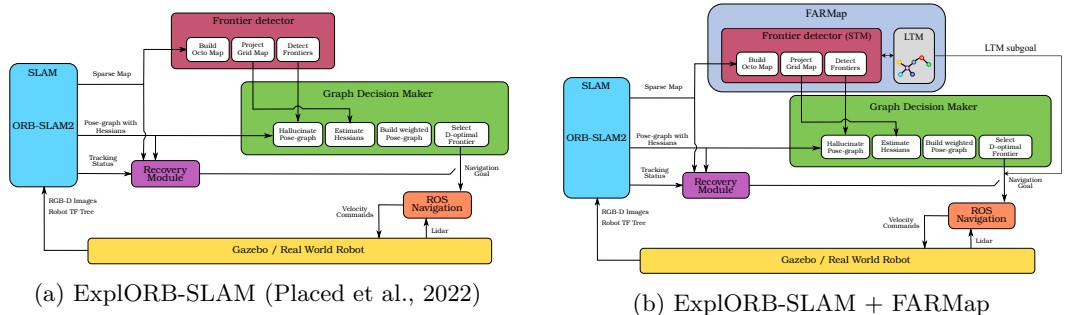

(a) ExplORB-SLAM (Placed et al., 2022)

(b) ExplORB-SLAM + FARMap

Figure 12: (a) The schematic diagram of ExplORB-SLAM from Placed et al. (2022). (b) An example of FARMap integrated ExplORB-SLAM. FARMap encapsulates 'Frontier Detector' and LTM subgoal can substitute 'Navigation Goal' from 'Graph Decision Maker.'

