# OpenReview forum: "Grid Cell-Inspired Fragmentation and Recall for Efficient Map Building"
_TMLR — Accepted by TMLR_

### Review · Reviewer_VhYd · 2024-05-17

**Summary Of Contributions:**

In this paper, the authors introduce a neuroscience-inspired algorithm to build maps locally by identifying points of fragment. New map formation occurs when high surprisal events occur. The algorithms seems to achieve good performance when compared to classical algorithms. Moreover, the authors state that their agents capture animal behaviour. The model seems to achieve promising results, but there are a few unclear points that should be addressed.

**Audience:**

Yes

**Broader Impact Concerns:**

No concerns.

**Claims And Evidence:**

No

**Requested Changes:**

See above in weaknesses a number of major points that should ideally be addressed.

Minor ones:
1. Why multi-coloured grid maps is utilised? Are the colours necessary, what happens if walls are kept with the same colour, can the model still detect surprisal events?
2. Why use custom maze and not an existing environment such as Minigrid gym (Chevalier-Boisvert, CoRR, 2023)?
3. How well does it generalise? Try in a new environment with similar condition (observation) but different layout.
4. “However, FARMap divides space based on properties of the space (how predictable the space is based on the local map or model), and does so in an online manner using surprisal.” What happens if the environment is much more simple and uniform.
5. Fig 5 unclear as why fracture points are in certain positions (right after the turn and not before as both events look similar in term of ‘surprisal’). Please elaborate on this more.
6. The definition of memory usage as given by local map size/ environment size is not clear. Considering local map is always W x H, shouldn’t the memory remain fixed? But the figures shows some variation as number of steps increases. As described here, it seems more like it is the observation size; how does it related to short-term memory (STM) and longe-term memory (LTM) and their respective sizes?

**Strengths And Weaknesses:**

Strengths:

The algorithm introduced outperforms existing mapping algorithms while being more efficient.

Comparison with neuroscience observations.

Several aspects of the same principle are explored and there is a good effort in explaining the algorithm.


Weaknesses:

1. The results need better quantification. For example, in Figure. 4, its not clear how well is the model able to reconstruct the original environment across the different sizes. This could be easily calculated by comparing reconstruction with original in a final panel e. And maybe compare here with a random agent and other existing methods?

2. Similar to above, the same problem occurs in figure 5. Can you quantify this somehow? Maybe using a metric that has been used in the original neuroscience papers. This figure is a bit unclear. Are you simply saying that your algorithm can identify the fragment points? Is that what is claimed in neuroscience? Or rather that the grid representations change after these points? If the later, then its unclear how the model is related to experiments. Accordingly, the 5.1 section is super short, needs more explanation. This needs more work as its one of the key points made in the abstract.

3. Related to the point above. More discussion related to the neuroscience literature would be helpful. For example, recent work has shown that RL agents trained to navigate partially observable environments better explain both neural and behavioural data in the hippocampus with good generalisation properties (Pedamonti et al. 2023 bioRxiv: https://doi.org/10.1101/2023.11.09.565503). Given that you also rely on partial observability, it would be important/interesting to discuss the relationship with this other work. See also more relevant papers within that paper.

4. In figure 6, these results should be shown with error bars/shaded areas, to quantify the variability in the models across different initial conditions. This would help the reader see how significant the results are. Similarly in Fig. 7, can you provide the linear correlation coefficients and respective p-values?

---

### Review · Reviewer_9Lry · 2024-05-22

**Summary Of Contributions:**

This paper proposed an efficient active map building method for mobile agents inspired by the remapping/fragmentation of grid cells. The proposed method builds new local maps based on surprisal signals and performs exploration in existing local maps based on the explored rate. The method is rule-based and heuristic-based. Since only the local map and fracture points of the local maps are stored in memory, the method is more memory-efficient than existing active mapping methods. The authors performed experiments in simulated environments, including randomly generated 2D grid environments and existing 3D room environments. The results show the proposed method can achieve a better mapping rate by using less memory and computation time.

**Audience:**

Yes

**Broader Impact Concerns:**

There is no broader impact concerns for this paper.

**Claims And Evidence:**

No

**Requested Changes:**

1. Recommend to move Section 3.7 to the beginning of the Method section. It will help the reader better understand the proposed method.

2. If we read the paper from the computational neuroscience perspective, the readers would like to learn insights into experimental neuroscience observations. Are there any new insights from the experiments and results of this paper that can better explain the underlying mechanism of neuroscience observations? For example, the authors can perform additional ablation studies and examine if the simulated result can show a similar effect as biological ablation studies.

3. The agent only recalls existing maps if it's on a fracture point. How to make sure the agent will always move on top of a fracture point when a change of a map is needed? Will the fracture points be very dense and cost a lot of memory space?

4. The computation of the surprisal signal is fully based on a simple 2D pattern matching algorithm. Will this computation method generalize to real-world environments with more complex patterns and noises? If not, how will the authors deal with this problem?

5. The results shown in Figure 6 and Figure 7 seem to contradict each other. Figure 6 shows the memory costs of FARMap are significantly lower than Frontier. However, in Figure 7, the memory cost differences are much less. For smaller environments, it seems the differences are even statistically insignificant.

6. A similar contradiction is also observed in Table 1-3, as the memory reduction is not as significant as in Figure 6 (For the small environment).

7. Since the fragmentation threshold is the lower the better in the ablation study, why not use 1.0 in the main result for all the experiments?

8. What will happen if decreasing decay factor to less than 0.8?

9. The authors only compare with the Frontier method. However, the mapping method in robotics always works with SLAM. How does the method compare with state-of-the-art Active SLAM methods?

**Strengths And Weaknesses:**

Strength

1. It's interesting to see the paper shows similar results as in neuroscience experimental findings. This can potentially give explanations for the underlying mechanism of grid cells.

2. The authors show the proposed method achieves better exploration rates on large simulated environments compared to the Frontier-based method by using much less memory and computation time.

Weakness

1. The review has a concern for the targeted audience of the paper. The paper may gain interest from neuroscience and traditional mobile robot researchers. However, it lacks connections with general machine-learning research. Maybe more computational neuroscience or robotics-oriented journals will be a better venue for the work.

2. The proposed method is fully rule-based and heuristic-based. There can be many problems with such a system for robotics. First, the system will lack flexibility and adaptability as the method is confined to predefined rules. For example, it's questionable if the method can work in open environments without well-defined borders and with other types of sensors. Second, the complexity of managing and scaling the system increases significantly with the complexity of the environment. Additionally, they require substantial expert knowledge, are inefficient in unstructured environments, and have difficulty handling noisy or incomplete data.

3. Therefore, it is questionable if the proposed method can be applied to real-world robots and environments. Since the experiments performed are either too abstract (2D grid environment) or too simple (simple 3D environment with structured borders), it's hard to make the connection with real-world deployments.

---

### Review · Reviewer_r3yZ · 2024-05-23

**Summary Of Contributions:**

This paper proposes a novel approach to build map for an environment, which is inspired by the working mechanism of the grid-cells in brains. The work focuses on the fragmentation of grid cell map in compartmentalized spaces, based on which the concept of 'fragmentation and recall' is proposed for mapping large spaces.

**Audience:**

Yes

**Claims And Evidence:**

Yes

**Requested Changes:**

Please discuss the two weakness points above.

**Strengths And Weaknesses:**

Strengths:

1. Biological brains are always source of inspiration for artificial intelligence development, therefore this work is well motivated and interesting. I think the study in this work is beneficial for the community.

2. The experiments are thoroughly conducted from multiple perspectives. The proposed method significantly outperform the baseline (frontier).

weakness:
1. It seems that only one baseline (frontier) is adopted for comparison. Since this method was proposed in 1997, please discuss whether this represents the current best performance and whether there are newer baselines.

2. Although the inspiration from biological brains is good, it would be better to discuss more on the relationship between the proposed method and the other state-of-the-arts methods. E.g. the inefficiency of the existing methods, and why the proposed method can resolve the issues.

---

### Decision · Action_Editor_7iqC · 2024-06-28

**Recommendation:** Accept as is

**Comment:**

I recommend the paper be accepted, given that it meets the criteria. I am also recommending this article for the featured certification because it not only shows benefits in the robotics environments, but also reproduces some neuroscientific findings of where grid cell remapping occurs in the environment; thus, it highlights potentially interesting connections that could be of benefit to both fields — in addition to clearly presenting the method and results, and connecting well with the relevant literature (especially in the revised version).

**Audience:**

The reviewers agreed that the revised paper supports its claims and connects to the literature across the fields better. The paper seems potentially of interest to computational neuroscientists as well as roboticists; thus, the reviewers agree that the paper is of interest to some of TMLR's audience.

**Claims And Evidence:**

The reviewers agree that the revised version of the paper addresses their main concerns, and adequately support its claims by comparing to baselines, and demonstrating that the proposed approach complements the existing neural SLAM approaches.